# Computational analysis into the potential of azo dyes as a feedstock for actinorhodin biosynthesis in *Pseudomonas putida*

**Parsa Nayyara**[1,2]*, **Dani Permana**[3]*, **Riksfardini A. Ermawar**[4], **Ratih Fahayana**[1]

**1** Sekolah Menengah Atas Negeri (SMAN) 5 Surabaya, Jalan Kusuma Bangsa No. 21, Surabaya, Indonesia, **2** University of British Columbia, Vancouver, British Columbia, Canada, **3** Research Center for Genetic Engineering, The National Research and Innovation Agency of the Republic of Indonesia (Badan Riset dan Inovasi Nasional (BRIN)), Kawasan Sains dan Teknologi (KST) Dr. Ir. H. Soekarno, Jalan Raya Jakarta-Bogor, Cibinong, Bogor, Indonesia, **4** Research Center for Biomass and Bioproducts, The National Research and Innovation Agency of the Republic of Indonesia (BRIN), Kawasan Sains dan Teknologi (KST) Dr. Ir. H. Soekarno, Jalan Raya Jakarta-Bogor, Cibinong, Bogor, Indonesia

\* nayyara@student.ubc.ca (PN); dani008@brin.go.id (DP)

**Data Availability Statement:** All relevant data used and generated in this study have been made available in this paper and Supporting Information files.

## Abstract

Fermentation-based biosynthesis in synthetic biology relies heavily on sugar-derived feedstocks, a limited and carbon-intensive commodity. Unconventional feedstocks from less-noble sources such as waste are being utilized to produce high-value chemical products. Azo dyes, a major pollutant commonly discharged by food, textile, and pharmaceutical industries, present significant health and environmental risks. We explore the potential of engineering *Pseudomonas putida* KT2440 to utilize azo dyes as a substrate to produce a polyketide, actinorhodin (ACT). Using the constrained minimal cut sets (cMCS) approach, we identified metabolic interventions that optimize ACT biosynthesis and compare the growth-coupling solutions attainable on an azo dye compared to glucose. Our results predicted that azo dyes could perform better as a feedstock for ACT biosynthesis than glucose as it allowed growth-coupling regimes that are unfeasible with glucose and generated an 18.28% higher maximum ACT flux. By examining the flux distributions enabled in different carbon sources, we observed that carbon fluxes from aromatic compounds like azo dyes have a unique capability to leverage gluconeogenesis to support both growth and production of secondary metabolites that produce excess NADH. Carbon sources are commonly chosen based on the host organism, availability, cost, and environmental implications. We demonstrated that careful selection of carbon sources is also crucial to ensure that the resulting flux distribution is suitable for further metabolic engineering of microbial cell factories.

## Introduction

Polyketides and their derivatives represent a clinically important group of secondary bioactive compounds comprising a range of FDA-approved antibiotics, immunosuppressants, and anti-cancers, making over 20B USD/year in the market [1] Due to its structurally diverse nature,

**Funding:** The first author is supported by the Indonesia Endowment Fund for Education / Lembaga Pengelola Dana Pendidikan (LPDP) of the Republic of Indonesia and the Education Financing Service Center / Pusat Layanan Pembiayaan Pendidikan (PUSLAPDIK) at the Ministry of Education, Culture, Research and Technology of the Republic of Indonesia under Beasiswa Indonesia Maju (BIM). We thank LPDP and PUSLAPDIK for funding the publication fees for this study. The funders had no role in study design, data collection and analysis, decision to publish, or preparation of the manuscript.

**Competing interests:** The authors have declared that no competing interests exist.

**Abbreviations:** ACT, Actinorhodin; MR, Methyl Red; GLC, Glucose; FBA, Flux Balance Analysis; FVA, Flux Variability Analysis; FSEOF, Flux Scanning based on Enforced Objective Flux; ***BiGG Reaction IDs*** PDH, Pyruvate Dehydrogenase; ACS, Acetyl-CoA Synthetase; ACALD, Acetaldehyde Dehydrogenase; 3OXCOAT, 3-oxoadipyl-CoA Thiolase; DMALRED, Malate Oxidoreductase; MDH, Malate Dehydrogenase; PC, Pyruvate Carboxylase; ADCL, 4-aminobenzoate Synthetase; MCITL2, Methylisocitrate Lyase; HOPNTAL, : 4-hydroxy-2-oxopentanoate Aldolase; EDD, 6-phosphogluconate Dehydratase; EDA, 2-dehydro-3-deoxy-phosphogluconate Aldolase; ENO, Enolase; AKGDa, Oxoglutarate Dehydrogenase; ICL, Isocitrate Lyase; PGL, 6-phosphogluconolactonase; PYK, Pyruvate Kinase; PPS, Phosphoenolpyruvate Synthase; PPCK, Phosphoenolpyruvate Carboxykinase; HSTPT, Histidinol-phosphate Transaminase; ILETA, Isoleucine Transaminase; ACOTA, Acetylornithine Transaminase; ALATA_L, L-alanine Transaminase; PSERT, Phosphoserine Transaminase; ASPTA, Aspartate Transaminase; ICDHyr, Isocitrate Dehydrogenase; SUCDi, Succinate Dehydrogenase; ARGSL, Argininosuccinate Lyase; ADSL, Adenylsuccinate Lyase; FE3DCITR3, Ferric Dicitrate Reductase; CS, Citrate Synthase; G3PD2, Glycerol-3-phosphate Dehydrogenase; TPI, Triose-phosphate Isomerase; FBA, Fructose-bisphosphate Aldolase; TKT1, TKT2: Transketolase; TRPS1, Tryptophan Synthase; TALA, Transaldolase; GAPD, Glyceraldehyde-3-phosphate Dehydrogenase; PGI, Glucose-6-phosphate Isomerase; PGK, Phosphoglycerate Kinase; PGM, Phosphoglycerate Mutase; PGL, 6-phosphogluconolactonase; GLYCtpp, Periplasmic Glycerol Transport; ALCD19, Alcohol Dehydrogenase; UGLYCH, Ureidoglycolate Hydrolase; GLYO1, Glycine Oxidase; GLYCTO1, Glycolate Oxidase; GND, Phosphogluconate Dehydrogenase; RPE, Ribulose-5-phosphate-3-

polyketides have also been utilized as the industrial building blocks for other specialized chemicals, such as pesticides [2] and biodegradable textile dyes [3]. Yet, industrial production of polyketide-derived chemicals is heavily reliant on large volumes of sugar feedstocks [4,5] that not only require vast area of land but are also energy intensive and release substantial amounts of $CO_2$ [4–6]. This current unsustainable method of manufacturing polyketides evokes the need to explore novel and non-conventional substrates [4,6–8], especially those originating from waste or biomass.

One abundant group of waste that has not been explored thus far as a feedstock for biomanufacturing are azo dyes. Azo dyes are xenobiotic compounds that account for roughly 70% of synthetic commercial dyes available today, which include a variety of dyes present in food, cosmetics, paper, therapeutics, and textiles [9]. Unfortunately, during many industrial processes, only 10% of the azo dyes applied are bound permanently to the colored object [10], while the remaining majority is released as hazardous effluents [9,11]. Physicochemical treatment or bioremediation using microbes is usually used to break down their (-N = N-) azo bond [10,12]. Simply breaking down these azo bonds without further processing releases aromatic amines that are highly toxic and carcinogenic [13]. Given the ability of many bacterial genera to catabolize aromatic compounds [14,15], by engineering bacterial metabolism, azo dyes and their aromatic amine by-product can be valorized to produce a heterologous compound such as polyketides.

Here, we explore the possibilities of utilizing and optimizing azo dyes as an alternative substrate for heterologous polyketide biosynthesis in *Pseudomonas putida* KT2440 through constraint-based genome-scale modeling. As a proof of concept, we focused on the synthesis of actinorhodin (ACT), which is the most extensively studied polyketide and is a model compound for understanding the genetics and metabolic mechanisms behind polyketide biosynthesis [16]. Successful implementation of our design for ACT production can suggest that the system applies to other industrially valuable polyketides. *P. putida*'s high GC content [17] could allow effective heterologous expression of GC-rich polyketide synthases (PKSs), including the ACT PKS from *Streptomyces coelicolor* [16,18]. Recent studies have demonstrated heterologous expression of PKSs in *P. putida* KT2440 with varying performances, either on par or better than the native host [18–20] and some inferior to other production hosts [21].

Additionally, *P. putida* becomes a suitable chassis for this biological system due to its high tolerance to chemical stress, ability to produce an azoreductase, and vast catabolic routes for aromatic compounds [14,17,22,23]. Biodegradation of simple azo dyes such as methyl red (MR) has previously been observed in *P. putida* MET94 [24,25] and *P. putida* OsEnB_HZB_G20 [26]. *P. putida* MET94 was selected from 48 bacterial strains for its ability to decolorize a range of common azo dyes to a higher extent compared to other strains [25]. Although specific growth rates decreased as MR concentration increased (indicating some level of toxicity), *P. putida* MET94 decolorized 90% of the dye under 24 h when grown in LB-rich media supplemented with 150 μM of the dye [25]. The azoreductase (PpAzoR) found in *P. putida* MET94 showed a 99% sequence similarity with an azoreductase found in *P. putida* KT2440 [24].

We employed constrained MCS (cMCS) [27,28] on the *P. putida* KT2440 iJN1462 model to identify knockout targets that enable growth-coupling of ACT from azo dyes. Recently, the use of cMCS for growth-coupling of a heterologous product has been performed and tested experimentally for the production of indigoidine from glucose in *P. putida* KT2440 [29]. Banerjee *et al.* [29] have also explored if the cMCS-suggested design for glucose is also applicable with other *P. putida* native substrates (*p*-coumarate and lysine), but it was found that growth-coupling was not maintained in any of these substrates. From this result, it appears that cMCS suggested designs are specific to the carbon source chosen. Another study by Kamp and Klamt

epimerase; NFORGLUAH2, N-formyl-L-glutamate Amidohydrolase; PTRCTA, Putrescine Transaminase; ABTA, : 4-aminobutyrate Transaminase; IG3PS, Imidazole-glycerol-3-phosphate Synthase; GLUDxi, Glutamate Dehydrogenase; ASNS1, Asparagine Synthase; GMPS2, GMP Synthase; PRFGS, Phosphoribosylformylglycinamidine Synthase; GLUPRT, Glutamine Phosphoribosyldiphosphate Amidotransferase; GLUDy, Glutamate Dehydrogenase; PSP_L, Phosphoserine Phosphatase; LLEUDr, L-Leucine Dehydrogenase; LEUTA, Leucine Transaminase.

[30] has found that growth-coupling was possible for almost all native metabolites in five production hosts with glucose as substrate, and expected that for some metabolites unfeasible to couple, changing carbon substrate may make growth-coupling feasible; however, this prediction was not tested further. Based on these studies and with the growing interest to transition from sugar-based substrates, we were interested to see if there are any differences in the cMCS solution space for a heterologous product when using glucose versus a non-sugar-based substrate, especially for specialized products like actinorhodin that are difficult to growth-couple. So far, the cMCS approach has not been used to optimize the utilization of non-traditional aromatic carbon sources like azo dyes to produce a heterologous product.

In metabolic engineering, carbon substrates, often mixed, are typically selected based on abundance, accessibility, cost, or the host organism's catabolic capabilities [4,8,18,29,31–34], and not based on the flux spaces and distributions needed by the metabolite of interest. Our computational analyses show that utilizing different carbon sources alters the growth-coupling solution spaces for ACT, i.e., some strain designs that optimize ACT production become feasible in one carbon source but not the other. The selection of carbon sources then becomes a crucial variable to consider in metabolic engineering for bioproduction. Moreover, models, synthetic pathways, and gene knockout suggestions generated in this work provide testable strain designs for a more circular and sustainable polyketide production system from azo dyes in *P. putida* KT2440.

## Material and methods

### Tools and instruments

Computations were performed using Python 3.8.5, Conda 4.9.2, on Jupyter Notebook 6.4.0 with CPLEX solvers. Genome-scale model analyses were carried out using the COBRApy Toolbox [35] and the StrainDesign [28] Python Package. The algorithms OptKnock [36], OptGene [37], and cMCS [38] were employed to examine genetic manipulation strategies for optimization. Escher [39] was used for pathway visualization purposes.

### Data sources

All data used for genome-scale model analyses were taken from public databases, namely MetaCyc [40], BioCyc [41], NCBI GenBank [42], UniProt [43], Kyoto Encyclopedia of Genes and Genomes (KEGG) [44], BiGG Models [45], and existing literature.

### Genome-scale model

The latest metabolic reconstruction of *P. putida* KT2440 iJN1462 [23] was the basis of the model used for genome-scale model analyses, with several changes to reaction contents (Table 1). This modified model, designated as iJN1462a, was used to simulate a wild-type *P. putida*.

**Table 1. Modifications made in iJN1462a (excluding boundary reactions).**

| Reaction Added | Reaction Description | Reason |
|---|---|---|
| AzoR_MR | Methyl Red [c] + 2 H+ [c] + 2.0 NADH [c] —> N,N-dimethyl-4-phenylenediamine [c] + Anthranilate [c] + 2.0 NAD+ [c] | Discarded in the original model because its role in the wild-type metabolism was unknown; re-added to find a synthetic pathway that would make this reaction metabolically essential |
| MRt1 | Methyl Red [e] < = > Methyl Red [p] | Enables MR transport from the extracellular compartment to the periplasm |
| MRt2 | Methyl Red [p] + H+ [p] < = > Methyl Red [c] + H+ [c] | Enables MR transport from the periplasm to the cytosol |
| EX_MR | Methyl Red [e] —> | MR exchange reaction |

To simulate the engineered *P. putida* harboring the ACT gene cluster (iJN1462b), 22 essential reactions for ACT biosynthesis were added from iAA1259 [46] into iJN1462a. iJN1462c (**S1 File**) was the model containing the complete heterologous pathway that enables azo dye conversion to ACT. The iAA1259 model representing wild-type *S. coelicolor* was also used for comparison in flux balance analysis (FBA) or flux variability analysis (FVA). All models only include reactions characterized in BiGG, BioModels, KEGG, or MetaCyc.

### *In silico* mediums and growth conditions

Uptake rates for all nutrients were set according to the *in silico* M9 minimal medium as specified by Nogales *et al.* [23], except for MR because its specific steady-state uptake rate in *P. putida* had not been tested experimentally before. Thus, because we aim to analyze the maximum theoretical capability of iJN1462c to consume MR in this study, the MR uptake rate in the medium, if any, was set to 1000 mmol gDW$^{-1}$ h$^{-1}$. The value 1000 was chosen to simulate an extreme case in which MR uptake was not constrained and the cell was free to consume as much MR as possible under the constraints of the model. We set the $O_2$ uptake rate to 30 mmol gDW$^{-1}$ h$^{-1}$ as specified by Nogales *et al.* [23] to simulate aerobic conditions. Carbon sources were varied as needed, using either GLC, MR, GLC + MR, or other aromatic compounds.

### Pathway design process

We used an FBA-based, model gap-filling method to construct our MR-to-ACT pathway since most of the metabolic framework is already native to *P. putida*. We started by defining the target metabolite, host, and growth medium, then ran FBA with ACTt as the objective and MR as the sole carbon source. If FBA returns an optimized value of the objective function, then a pathway to synthesize ACT from MR exists. The model is said to be unfeasible when no pathway was found to connect the two nodes. Initially, iJN1462b returned unfeasible on MR only and with ACTt as objective. To fill incomplete gaps in the pathway, we mapped the model in Escher to identify where the gaps lie and searched for compatible reactions in MetaCyc, NCBI GenBank, and UniProt to fill in missing links. If the model becomes feasible upon addition of a heterologous reaction, then that reaction is added to the model as part of the MR to ACT pathway. Otherwise, the process is repeated until it returns feasible. For our pathway, only one reaction, AnthDO, was needed to connect missing links in iJN1462b. The model containing the pathway that enabled MR conversion to ACT was designated iJN1462c (**S1 File**).

### Calculating flux-sum

We retrieved the fluxes of reactions producing and consuming relevant cofactors from FVA (**S2 File**). Since the sum of producing and consuming reactions in steady-state models is always equal to zero, we used the concept of flux-sum to measure cofactor regeneration rates. Flux-sum ($\Phi_i$) was determined using the following formula [32],

$$\Phi_i = 0.5 \sum_j |S_{ij} v_j| \tag{1}$$

where

$\Phi_i$: Flux-sum of metabolite $I$;
$S_{ij}$: Stoichiometric coefficient of metabolite $i$ in reaction $j$; and
$v_j$: Flux of reaction $j$.

## Model preprocessing for cMCS calculation

Prior to cMCS computations, iJN1462c was reduced into a set of target reactions and genes that are biologically valid to be knocked out; this was different for each medium with different carbon source. In addition, pre-processing was also useful to reduce hours of MCS runtime significantly. Reactions and the associated genes eliminated from iJN462c were:

a. Blocked reactions (reactions that were unable to carry flux);

b. Essential reactions (reactions that would result in a lethal phenotype i.e., no biomass formation if removed);

c. Non-gene-associated reactions (reactions that are not encoded by a gene or whose gene association is unknown);

d. Transport and peripheral reactions (these reactions are very difficult to manipulate experimentally);

e. Boundary reactions (exchange, sink, and demand reactions in the model).

## cMCS computations

MCS was computed using the StrainDesign package [28] in Python. We specified any ACT flux under the tested thresholds (0, 1, 5, 10, 30, and 50% of optimum flux) as the target or undesired region. Simultaneously, we set any biomass flux above the thresholds being tested as the protected or desired region. We started by computing MCS with the most relaxed constraints, i.e., the lowest thresholds, then gradually increasing them until no solutions were found. Only reactions in the pre-processed model are considered valid knockout suggestions. Each MCS computation was allowed to search within a time limit of 300 seconds.

## Pathway utilization analysis

To examine which pathways were activated in each strain design when grown on different carbon substrates, 23 representative metabolites from key pathways in the cell's carbon metabolism were chosen. We then calculated the total flux producing these metabolites to indicate whether a pathway is utilized in a given strain design. We also observed which reactions produced these key metabolites to determine directionality and ascertain if their producing pathways have been rewired differently. The analysis was performed under biomass as the objective to simulate a real-life scenario where the cell would be optimizing for growth. For comparison, iJN1462c as control was analyzed under both biomass and ACT production as an objective to examine where fluxes need to be channeled to support ACT or biomass production.

   Acetyl-CoA, oxaloacetate, pyruvate, and phosphoenolpyruvate were chosen to represent the central carbon metabolism and elucidate activation of glycolysis or gluconeogenesis; a-ketoglutarate, fumarate, and citrate reach present the TCA cycle; fructose-1,6-biphosphate represents the EMP pathway; glucose-6-phosphate, glyceraldehyde-3-phosphate, and 3-phosphoglycerate represent glycolysis; 6-phosphogluconate and 2-dehydro-3-deoxy-D-gluconate-6-phosphate represent the ED pathway; glyoxylate represents the glyoxylate shunt; acetaldehyde elucidates aromatic catabolism; ribulose-5-phosphate and xylulose-5-phosphate represent the PP pathway and indicates whether it is oxidative or non-oxidative; dihydroxyacetone phosphate and glycerate elucidates gluconeogenesis; and glutamate, serine, aspartate, and leucine is chosen to observe amino acid biosynthesis.

## Strain design simulations

Each reaction knockout set was verified with standard FBA analysis under biomass as the objective and FVA was implemented to ensure guaranteed ACT production even when cells do not grow (i.e., minimum ACT yield is not 0). For each reaction knockout target, genes encoding the associated enzymes were identified. To verify gene-protein-reaction (GPR) associations, we performed another knockout simulation using only gene knockouts and observed if the gene knockout agreed with reaction knockouts (**S3 File**). For reactions catalyzed by isoenzymes, genes encoding all known isoenzymes are knocked out. For reactions catalyzed by enzymes with subunits or encoded by multiple genes, only one gene is knocked-out since the associated enzyme would not be functional without one of the subunits. Knocking out genes controlling more than one reaction is avoided. Only strain designs where the gene knockout agrees with the suggested reaction knockouts are evaluated in the next step. The production envelope of each strain design was plotted to examine the altered production space (**S4 File**).

## Product yield and BPCY calculation

Strain designs were evaluated based on their knockout size (>25 gene knockouts are undesirable), product yield (at the maximum growth rate and the minimum guaranteed ACT yield), and biomass-product coupled yield (BPCY) [31] calculated using the equations shown below.

$$Product\ Yield = \frac{production\ rate\ _{actinorhodin}\ (mmol\ gDW^{-1}h^{-1})}{consumption\ rate\ _{carbon\ source}\ (mmol\ gDW^{-1}h^{-1})} \qquad (2)$$

$$BPCY = Product\ Yield \times growth\ rate\ (mmol\ gDW^{-1}h^{-1}) \qquad (3)$$

## Results and discussion

### Enabling azo dye conversion to polyketides in *P. putida* KT2440

The primary goal of this work was to explore the potential of using azo dyes as a feedstock for ACT biosynthesis. Specifically, we employed model-driven analysis to find possible regimes that allow growth-coupling of ACT biosynthesis from the model azo dye methyl red (MR). We based our analysis on the latest genome-scale metabolic reconstruction of *P. putida* KT2440, iJN1462 [23]. The monoazo MR was chosen as a model compound for azo dyes since it is the only azo dye whose reaction with *P. putida* KT2440*'s* azoreductase has been well-characterized in MetaCyc [24,25,40]. Consequently, anthranilate, the reductive cleavage product of MR, was used as the model aromatic amine. Analyses were done under aerobic conditions using the *in silico* M9 minimal medium as described by Nogales *et al.* [23] with MR as the sole carbon source.

To enable heterologous production of the target compound, ACT, the iJN1462 model was then modified to include the ACT biosynthetic pathway from the validated iAA1259 genome-scale model. The iAA1259 model is the latest reconstruction of *S. coelicolor* that most accurately predicts ACT production fluxes in the stationary phase [46]. This modified version of the model, denoted iJN1462b, was still unable to grow and produce ACT from MR alone due to missing reactions between the anthranilate node and the central carbon metabolism. Through manual gap-filling based on Flux Balance Analysis (FBA), we identified a combination of native and non-native reactions (**S5 File**) defined in MetaCyc and KEGG that enables the iJN1462 *P. putida* model to use MR as a feedstock for ACT biosynthesis (**Fig 1**). The metabolic model capable of MR-to-ACT conversion was designated as iJN1462c (**S1 File**).

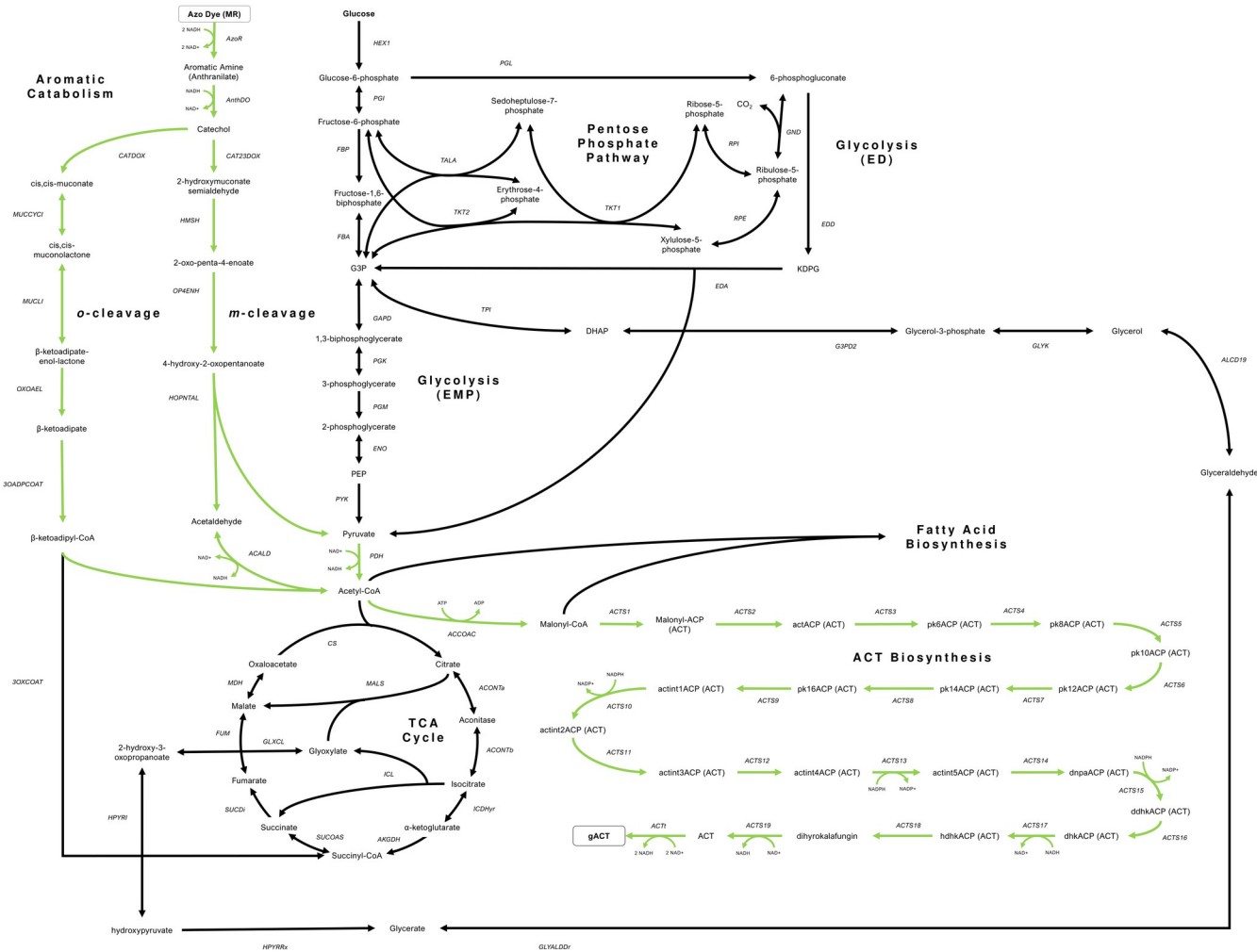

**Fig 1. Simplified map of the catechol degradation pathways, ACT biosynthesis, and central carbon metabolism in iJN1462c.** The engineered pathway connecting MR degradation and ACT biosynthesis is highlighted in green.

Adding a heterologous anthranilate dioxygenase made it possible for *P. putida* KT2440 to degrade MR and anthranilate through the meta- or ortho-cleavage pathway of catechol. The catechol branch is a common route for the aerobic catabolism of aromatic amines in bacteria, which usually begins with the formation of catechol from the aromatic amine by a dioxygenase [14]. In our engineered pathway (**Fig 1**), MR was first reduced by *P. putida* KT2440's FMN-dependent azoreductase into its aromatic amine, anthranilate. While *P. putida* is known for its ability to degrade aromatics such as toluene, benzoate, and naphthalene [15], no study thus far has reported the presence of a specific dioxygenase for anthranilate degradation in this strain. To complete this reaction gap in the synthetic pathway, a non-native anthranilate-1,2-dioxygenase reaction (AnthDO) from *Acinetobacter* sp. [47] encoded by the *ant*ABC operon was added. It is also worth noting that several studies [47,48] have shown similarities between this anthranilate-1,2-dioxygenase and toluene-1,2-dioxygenase encoded by the *xyl*XYZ operon in *P. putida* TOL plasmid pWW0. The similarity in sequence and function of the two enzymes suggests that toluene-1,2-dioxygenase might also be compatible with anthranilate, hence there is no need to heterologously express the *ant*ABC operon. Nonetheless, further experimental

work needs to be done to confirm this hypothesis; thus, *ant*ABC was still included in our pathway. It is important to note that although the iJN1462 model contains reactions encoded in the TOL plasmid, including the toluene-1,2-dioxygenase and the meta-cleavage pathway, *P. putida* KT2440 is a cured strain derived from *P. putida* mt-2 and does not possess the plasmid [23,49]. For experimental demonstration of the designs presented in this study, one would have to express the TOL plasmid in KT2440 or use the mt-2 parental strain.

*P. putida* also possess another pathway that converts anthranilate to tryptophan, which could be an alternative to the pathway via AnthDO. However, the conversion of anthranilate to tryptophan requires a supply of L-serine and 5-Phospho-alpha-D-ribose-1-diphosphate, which prevents this pathway from being the main route for anthranilate degradation. Without an external supply of either compounds or an additional carbon source to support biosynthesis, iJN1462b without AnthDO cannot grow or produce ACT from MR only (see supplementary notebook at the GitHub repository). Our supplementary tests also showed that although anthranilate is converted into tryptophan in the presence of an additional carbon source, the flux through the anthranilate-to-tryptophan pathway is low compared to the flux from the additional carbon source, indicating that anthranilate was not the main carbon source for growth. Similarly, in the presence of AnthDO with MR as sole carbon source, the flux through the tryptophan pathway is still very low compared to the flux through AnthDO. Therefore, despite the presence of an alternate anthranilate degradation pathway, AnthDO is still the preferred route to channel carbon flux from anthranilate to the central metabolic pathways.

Catechol produced via AnthDO is then catabolized into acetyl-CoA either via the meta-cleavage or ortho-cleavage pathway, connecting dye degradation to the central carbon metabolism. Funneling the carbon fluxes from azo dye degradation into the central acetyl-CoA node opens various possibilities for biomanufacturing since acetyl-CoA is the precursor to a wide range of commodity and specialized chemicals, including but not limited to polyketides, polyhydroxyalkanoates, isoprenoids, and fatty acids [33]. In ACT biosynthesis, units of acetyl-CoA and malonyl-CoA both form the main polyketide backbone via reactions catalyzed by the non-native type-II PKS from *S. coelicolor* [16,46]. Following the formation of the backbone are modifications by aromatases, cyclases, and dehydrogenases encoded in the ACT biosynthetic gene cluster [16,46].

## Evaluation of the production space via genome-scale modelling

To understand the theoretical production potential and limitations of the synthetic pathway, we conducted Flux Variability Analysis (FVA), an extension of FBA, on iJN1462c and mapped the feasible flux ranges in production envelopes. To assess the differences in the production space when MR or sugar-based substrate such as glucose (GLC) was used as feedstock, FVA was performed in the *in silico* M9 Minimal Medium with three carbon source variations: GLC only, MR only, and GLC + MR. Production envelopes depicting the relationships between optimizing cell growth (BIOMASS_KT2440_WT3), MR consumption (MRt1), and ACT production (ACTt) in these mediums are shown in **Fig 2**. To compare the ACT production space in *S. coelicolor* and the engineered *P. putida* KT2440, an ACTt vs biomass production envelope generated by iAA1259 in its exponential phase using a specific *S. coelicolor* growth medium described by Amara *et al.* [46] was also investigated.

FVA revealed the following. First, MR degradation was growth-coupled (**Fig 2A** and **2B**), but it eventually competed with biomass production once it reached a certain MR uptake rate (9.2 mmol gDW$^{-1}$ h$^{-1}$ and 5.7 mmol gDW$^{-1}$ h$^{-1}$ in MR and GLC+MR, respectively). Theoretically, the cell was able to consume MR up to 54.7 mmol gDW$^{-1}$ h$^{-1}$ with MR as the sole carbon source and 85.5 mmol gDW$^{-1}$ h$^{-1}$ with GLC+MR (**Fig 2A** and **2B**); however, pushing MR

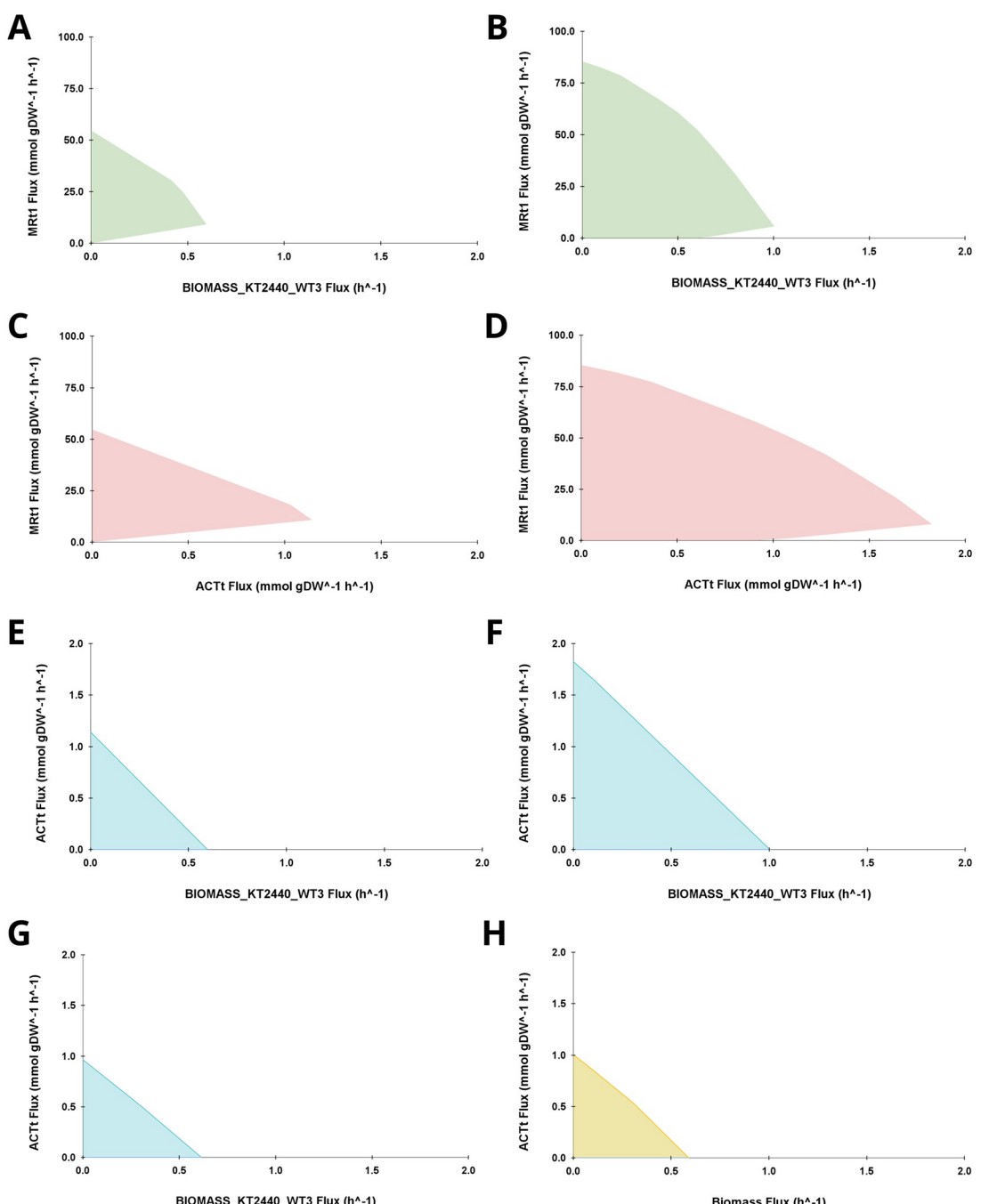

**Fig 2. Production envelopes mapping feasible flux ranges calculated through FVA.** (A) MRt1 vs BIOMASS_KT2440_WT3; MR. (B) MRt1 vs BIOMASS_KT2440_WT3; GLC+MR. (C) MRt1 vs ACTt; MR. (D) MRt1 vs ACTt; GLC+MR. (E) ACTt vs BIOMASS_KT2440_WT3; MR. (F) ACTt vs BIOMASS_KT2440_WT3; GLC+MR. (G) ACTt vs BIOMASS_KT2440_WT3; GLC. (H) ACTt vs Biomass; iAA1259 *S. coelicolor* model with GLC.

consumption above 9.2 and 5.7 mmol gDW$^{-1}$ h$^{-1}$ would decrease cell growth. By looking into cofactor consumption and regeneration in iJN1462c (**S2 File**), we found that the competition between MRt1 and BIOMASS_KT2440_WT3 was due to MR degradation requiring NADH as a redox cofactor, specifically for the reduction of MR into anthranilate (AzoR_MR) and then

into catechol (AnthDO). As more MR was consumed, more NADH flux was redirected from NADH dehydrogenase (NADH16pp) in the electron transport chain to MR degradation. With less NADH available for oxidative phosphorylation, less ATP was generated. The decrease in ATP flux negatively affected cell growth since BIOMASS_KT2440_WT3 was the main consumer of ATP when the cell's objective was optimizing biomass (**S2 File**).

Second, iJN1462c displayed an almost identical pattern of MR consumption when the objective was changed from BIOMASS_KT2440_WT3 to ACTt (**Fig 2C** and **2D**), where MR uptake was also coupled with ACT production up until a specific point. Again, this is because NADH was needed for both MR degradation and ATP synthesis. ACT biosynthesis necessitated malonyl-coa as a precursor, and malonyl-coa production from acetyl-coa required a considerable amount of ATP. Optimizing ACT production took up an MRt1 flux of 10.8 mmol gDW$^{-1}$ h$^{-1}$ and 8.0 mmol gDW$^{-1}$ h$^{-1}$ in Medium B and C, respectively, a slightly higher value than what could be achieved by optimizing BIOMASS_KT2440_WT3. We predicted that the higher MRt1 uptake rate attainable by optimizing ACT compared to optimizing growth, might be due to differences in how cofactors were regenerated when the cell objective was changed. Hence, we analyzed the turnover rates of NADH, NADPH, ATP, and ADP in iJN1462c by calculating their flux-sums [32] (**Fig 3**). As seen in **Fig 3**, optimizing ACT biosynthesis required less ATP supply than optimizing BIOMASS_KT2440_WT3 which explains why it was able to consume more MR. Moreover, ACT biosynthesis seems to be better than BIOMASS_KT2440_WT3 at regenerating NADH, thus helping to cover the loss of NADH flux in oxidative phosphorylation by MR degradation.

Third, because both BIOMASS_KT2440_WT3 and ACT biosynthesis required ATP and have a similar level of MRt1 flux demand, without any metabolic intervention it is impossible to optimize both cell growth and ACT production at the same time (**Fig 2E** and **2F**). The trade-off between ACT production and biomass was observed in all mediums regardless of the carbon source (**Fig 2E**, **2G**, and **2H**) as well as in *S. coelicolor* (**Fig 2H**).

Compared with *S. coelicolor*, iJN1462c had the potential to produce the same amount of ACT flux as *S. coelicolor* (~1.0 mmol gDW$^{-1}$ h$^{-1}$) when GLC was the sole carbon source (**Fig 2G** and **2H**). With MR as the carbon source, this value was increased by 18.28% to 1.1 mmol gDW$^{-1}$ h$^{-1}$ (**Fig 2C** and **2E**). Combining MR and GLC (**Fig 2B** and **2D**) provided an additional carbon flux to the system, enabling the cell to generate more BIO-MASS_KT2440_WT3 (1.0 h$^{-1}$) and ACTt flux (1.8 mmol gDW$^{-1}$ h$^{-1}$).

Regardless of the objective, adding GLC into the medium lowers the uptake rate at which MR can be consumed without any decrease in growth rate or ACT flux. This is because consuming GLC created more NADH demand in the system. In iJN1462c, the Entner-Doudoroff (ED) pathway was preferred to the Embden-Meyerhoff-Parnass (EMP) pathway to consume GLC, a typical occurrence in bacteria [50]. The ED pathway produced a net of 1 ATP and consumed a net of 1 ADP per GLC molecule. In contrast, no ADP was required, and no ATP was generated from MR degradation. With MR as a carbon source, ATP is produced by channeling carbon flux generated by consuming MR to the TCA cycle. Thus, a higher ATP turnover rate was observed in GLC-containing mediums because quick regeneration of ATP/ADP is needed (**Fig 3**). As larger ATP pools were needed to facilitate the high ATP turnover, more NADH flux was demanded for oxidative phosphorylation, causing the competition between MR degradation and ATP synthesis to happen at a lower MR uptake rate with GLC compared to without GLC. The cell would have been able to consume MR at a higher rate, without compromising growth or ACT flux, if GLC is not consumed at the same time.

By examining the production space of iJN1462c with FVA and analyzing cofactor distributions, we noticed that MR has the potential to be a better feedstock for ACT biosynthesis than GLC because (i) it was shown to produce a higher maximum theoretical ACTt flux and (ii) it

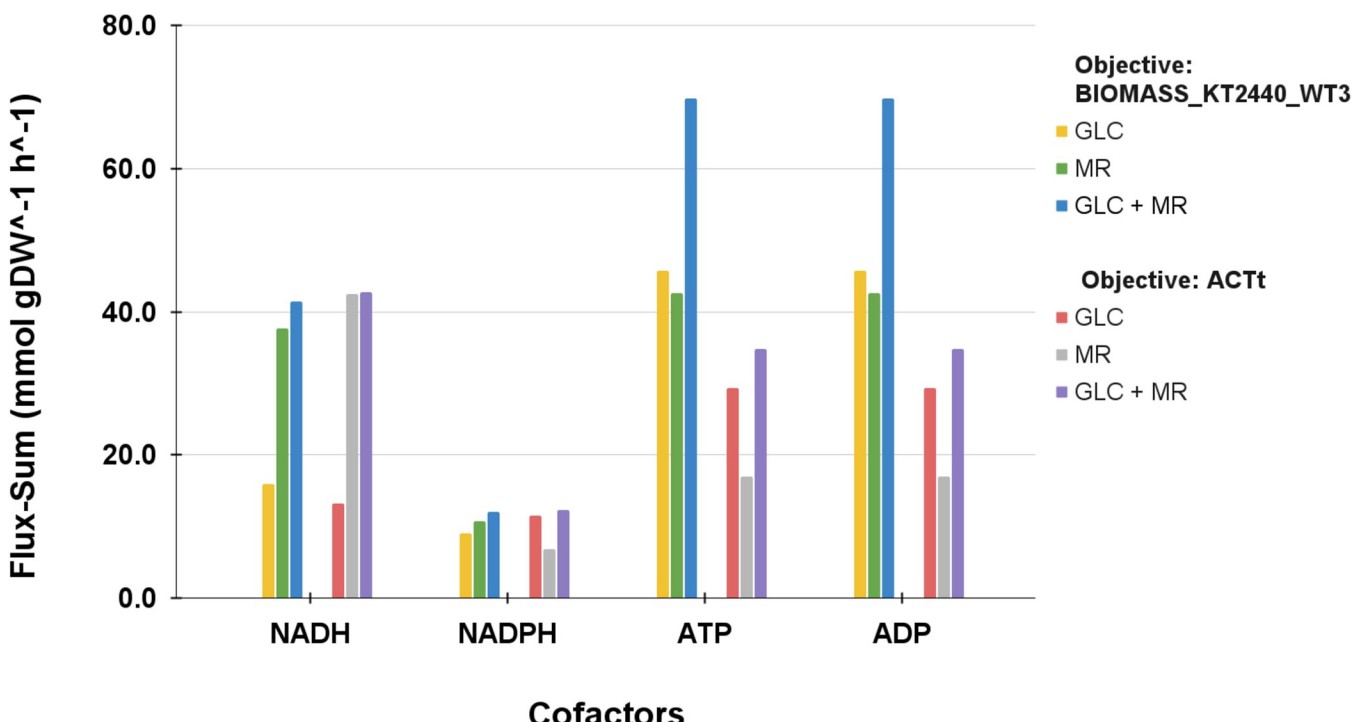

**Fig 3. Cofactor turnover rates in iJN1462c measured by their metabolite flux-sum.** Note that the difference in cell objective and carbon source affects cofactor flux-sum.

demands less ATP turnover to consume compared to GLC. Cofactor turnover rates, or flux-sum, is not to be mistaken with the ATP maintenance requirement which shows the minimum ATP flux needed for the cell to maintain non-growth associated cellular processes and is specific to each organism. For *P. putida*, the ATP maintenance requirement is 0.92 mmol ATP gDW$^{-1}$ h$^{-1}$ according to Ebert *et al.* [51]. By contrast, flux-sum reflects the amount of cofactor available in the system which varies according to the carbon source. It is also important to note that FVA and flux-sum analyses are based on linear, static, constraint-based modeling and do not account for factors such as gene regulations, reaction kinetics, and cellular stress responses that can influence MR uptake and subsequently ACT production from MR. Nevertheless, the analysis shown here reveals that mathematically, using MR as a carbon source allows a flux distribution that supports ACT biosynthesis more than GLC.

FVA has also shown that MRt1 and ACTt were greatly influenced by cofactor distribution and were not naturally growth coupled. Such poorly balanced cofactor pool and competition between biomass and the biosynthesis objective is a common encounter when engineering new synthetic pathways [34,52]. Thus, further metabolic interventions such as gene knockouts, overexpression, or downregulation are needed to optimize MR conversion into ACT in this engineered system.

### Identification of knockout targets for growth-coupling

A widely used approach to optimize a strain design and overcome the trade-off between biomass/growth and the biosynthesis objective is to enforce growth-coupling, which entails modifying the metabolic wiring of the organism such that the targeted biosynthesis product becomes an obligatory by-product of growth. Growth-coupling strategies can be identified either through FBA-based methods such as Optknock [36] and OptGene [37], or through the

enumeration of minimal cut sets (MCS) [38]. FBA-based methods use Mixed-Integer Linear Programming (Optknock) or heuristic evolutionary algorithms (OptGene) to identify knock-out targets that maximize the target objective within the solution space that maximizes bio-mass/growth, hence allowing growth-coupling. While these approaches are considerably robust in generating growth-coupled strain designs, the algorithm is structured around a nested bi-level optimization problem where the bioproduction objective is maximized subject to maximal biomass production [36].

Previously, we have attempted to use both Optknock and OptGene to generate knockout suggestions that optimize the MR to ACT pathway (**S4 File**). Solutions suggested by Optknock enabled very weak growth coupling (**S4 File**) and OptGene found no feasible knockout targets. Due to the strong trade-off between biomass and ACT production, it has likely been the case that ACT production is not feasible under optimum growth. However, a growth-coupled design could still be possible with non-optimum growth. Unlike FBA-based methods, MCS is not biased towards maximizing growth and allows for solutions that guarantee ACT production at submaximal biomass fluxes.

Here, we employed constrained MCS (cMCS) [27,28] to identify knockout targets that enable growth-coupling of ACT. iJN1462c was pre-processed prior to cMCS calculation to exclude all reactions that are invalid to be knocked out. This includes any essential and blocked reactions, non-gene-associated reactions, transport, peripheral, and boundary reactions. cMCS was calculated by systematically increasing the minimum theoretical ACT flux and the minimum desired biomass flux to 0, 1, 5, 10, 30, and 50% of the optimum. The computation was repeated in the three different carbon source compositions: GLC, MR, and GLC+MR. Reaction knockout sets found, and their associated genes are listed in **S3 File**.

## Using azo dye as carbon source facilitates growth-coupling

203 MCS solutions were found at different biomass and ACT flux thresholds in the MR media, 10 solutions in GLC+MR, and 0 solutions with GLC as the sole carbon source. Out of the 213 solutions found in total, only 4 strain designs had a gene knockout phenotype that agrees with the reaction knockout phenotype suggested by cMCS (**S3 File**). All four designs use MR as their sole carbon source. **Table 2** shows the growth-coupled strain designs found for ACT production from MR. The reaction knockouts suggested in context of key pathways in iJN1462c is illustrated in **Fig 4**.

**Table 2. cMCS-suggested strain designs.**

| Strain Design | % Minimum Biomass | % Minimum ACT | Reaction Knockouts | Gene Knockouts |
|---|---|---|---|---|
| 1 | 0 | 0 | MALS', 'FDH', 'SUCD4', 'ALDD2y', 'ALDD2x', 'SUCDi', 'GLYOX', 'NACODA', 'PGL' | PP_0356, PP_2183, PP_0489, PP_4203, PP_2589, PP_0545, PP_2694, PP_4190, PP_2492, PP_4144, PP_5186, PP_1023 |
| 2 | 1 | 0 | MALS', 'ALDD2y', 'MTHFC', 'ALDD2x', 'SUCDi', 'GLYOX', 'NACODA', 'ACACt2pp', 'RPE' | PP_0356, PP_2492, PP_2265, PP_1945, PP_2589, PP_0545, PP_2694, PP_4190, PP_4144, PP_5186, PP_0415, PP_3124 |
| 4 | 1 | 1 | SUCOAS', 'HIBDkt', 'PUTA3', 'ALDD2y', 'GLYOX', 'ACS2', 'PPS', 'ACONTb', 'GLYO1', 'PGL', 'ACACt2pp', 'MDH', 'ALDD2x', 'NTD11', 'FTHFD', 'PC', 'GARFT', 'ICL', 'IMPD', 'P5CD' | PP_4186, PP_4666, PP_4947, PP_2492, PP_4144, PP_2351, PP_2339, PP_2112, PP_2082, PP_1023, PP_0612, PP_3124, PP_0654, PP_2589, PP_0545, PP_2694, PP_1620, PP_0327, PP_1367, PP_1943, PP_1664, PP_5347, PP_4116, PP_1031, PP_4947 |
| 5 | 1 | 5 | ALDD2y', 'FTHFD', 'ALDD2x', 'MDH', 'MICITDr', 'GLYOX', 'ACACT11', 'MALS', 'PGL', 'ACONTa', 'NACODA' | PP_2492, PP_0327, PP_1367, PP_1943, PP_2589, PP_0545, PP_2694, PP_0654, PP_2336, PP_4144, PP_0356, PP_1023, PP_2112, PP_2339, PP_5186, PP_2137, PP_2215 |

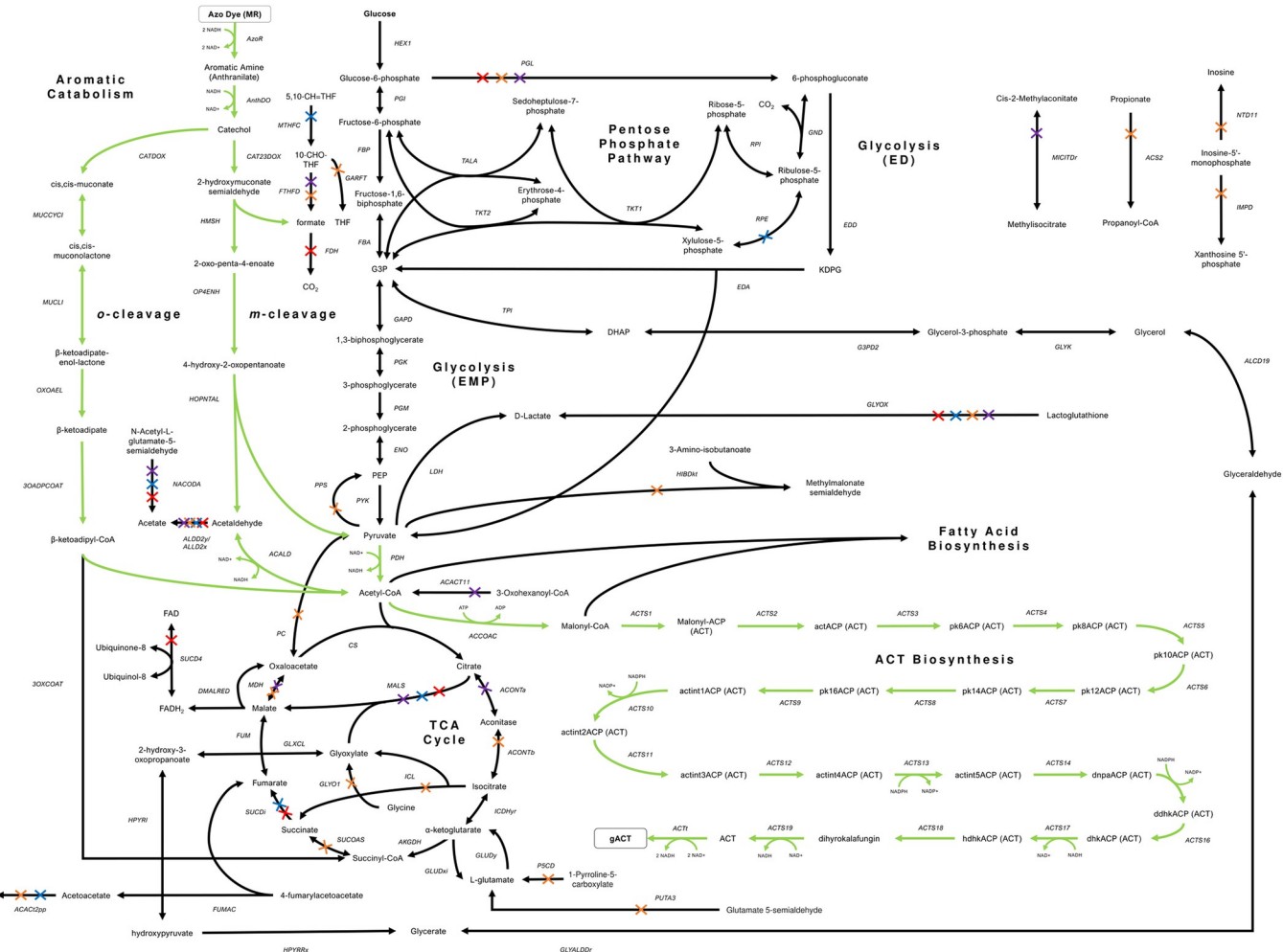

**Fig 4. Suggested reaction knockouts by cMCS.** The engineered pathway connecting MR degradation and ACT biosynthesis is highlighted in green. Knockouts in strain design 1: red cross; 2: blue cross; 4: orange cross; and 5: purple cross.

Our results highlight that the number of possible growth-coupling regimes and the minimum product yield allowable through growth-coupling is dependent on the type of substrate defined in the MCS computation. Using MR as a carbon source alters the solution space and enables growth-coupled designs that would not otherwise be possible with GLC. It is worth noting that growth-coupled designs for ACT were only found when MR is present as a carbon source, and none was found in the GLC-only medium.

*P. putida* has been found to segregate fluxes from different carbon substrates for use in specific carbon metabolic pathways when grown in mixed 1:1 glucose:benzoate medium [53]. Through $^{13}$C-labelling analysis, Kukurugya *et al.* showed that labeled glucose-derived carbon constitutes most of the metabolites involved in the upper glycolysis pathways (ED, EMP, and PP pathways), while the TCA cycle is primarily populated with benzoate-derived carbons. This means that in a mixed glucose:benzoate medium, carbon fluxes generated from GLC is used almost exclusively for generating cellular energy, while carbon from benzoate is used for biosynthesis. Moreover, they also observed a 10-fold increase in the acetyl-CoA pool in cells grown on glucose:benzoate compared to glucose only.

Benzoate and MR catabolic pathways both converge to the catechol node and enter the central carbon metabolism through acetyl-CoA, thus we suspect that the different MCS solution space created from using different carbon sources is in part due to this tendency for *P. putida* to direct carbon fluxes from glucose to the ED, EMP, and PP pathways, and MR-derived carbon fluxes into the TCA cycle and glyoxylate shunt. With GLC only, the cell can only draw acetyl-CoA fluxes from the glycolytic pathways. Meanwhile, with an aromatic carbon source, fluxes from the TCA cycle can be recycled back to the upper glycolytic pathways via gluconeogenesis, creating two channels of carbon flux for acetyl-CoA synthesis. Therefore, when the biosynthetic objective being optimized is a derivative of acetyl-CoA, an aromatic carbon source like benzoate and MR can facilitate a distribution of carbon fluxes that both supports cellular energy maintenance and the production of the metabolite of interest.

We tested this hypothesis in iJN1462c by looking at the fluxes channeled through several key metabolites that represent major pathways in *P. putida*'s carbon metabolism (**Fig 5**). Examining which metabolites are produced, and thus, which pathways are utilized or deactivated when the optimization objective is BIOMASS_KT2440_WT3 versus ACT, reveals where the carbon flux is needed to maximize growth or ACT production. Fluxes in the growth-coupled cMCS designs were also compared to see how fluxes can be redistributed to support both objectives.

Acetyl-CoA is the central node connecting most pathways in the central carbon metabolism and is the precursor for fatty acid and ACT biosynthesis. On GLC, acetyl-CoA is produced from pyruvate by PDH through the typical glycolysis route. When optimizing for growth on MR, the cell will produce acetyl-CoA via glycolysis (PDH), and the degradation pathways of catechol (**Fig 5**). Acetyl-CoA is also regenerated from acetate in iJN1462c grown on MR via acetyl-CoA synthetase (ACS). For maximum ACT production, a higher level of acetyl-CoA than iJN1462c is required. On GLC, this flux is generated from glycolysis, while on MR it is preferred that acetyl-CoA is channeled from glycolysis and the meta-cleavage pathway in a nearly 1:1 ratio. Interestingly, in growth-coupled designs, flux coming into the acetyl-CoA node was almost entirely drawn from the meta-cleavage pathway of catechol, except in strain design 4 which draws carbon flux from both glycolysis and the ortho-cleavage pathway. In strain designs 1, 2, and 5, instead of using PDH to convert pyruvate into acetyl-CoA, pyruvate is carboxylated into oxaloacetate (PC), which is then converted into phosphoenolpyruvate to redirect flux to produce glyceraldehyde-3-phosphate, via gluconeogenesis (**Fig 6**).

The activation of gluconeogenesis in strain designs 1, 2, and 5 is further confirmed by the level and sources of carbon flux found in the oxaloacetate, pyruvate, phosphoenolpyruvate, glyceraldehyde-3-phosphate (**Fig 5**), and the dihydroxyacetone phosphate nodes (**Fig 6**). In iJN1462c and strain design 4 grown on MR, oxaloacetate is generated through MDH in the TCA cycle, while in strain design 1, 2, and 5, it is generated from pyruvate by PC. In MR-grown cells, pyruvate is produced mostly from the aromatic catabolism pathway (HOPNTAL); but in strain design 4, flux from acetaldehyde is first directed to generate threonine and propanoyl-CoA before being converted into pyruvate via MCITL. On GLC, pyruvate is synthesized from glycolytic pathways, namely via EDA and PYK. For iJN1462c and the growth-coupled designs grown on MR, phosphoenolpyruvate is synthesized from pyruvate through PPCK, indicating a flux entering gluconeogenesis from the phosphoenolpyruvate node. In contrast, there is no production of phosphoenolpyruvate when ACT is maximized on MR. GLC-grown cells utilize ENO in the lower glycolysis pathway to produce phosphoenolpyruvate.

Flux directionality through the glyceraldehyde-3-phosphate (**Fig 5**), dihydroxyacetone phosphate, glycerate, and glyoxylate node (**Fig 6**) was also changed by changing cellular objectives and carbon sources. When growth is optimized on GLC, glyceraldehyde-3-phosphate is generated through the upper ED pathway and is then broken down through the glycolysis

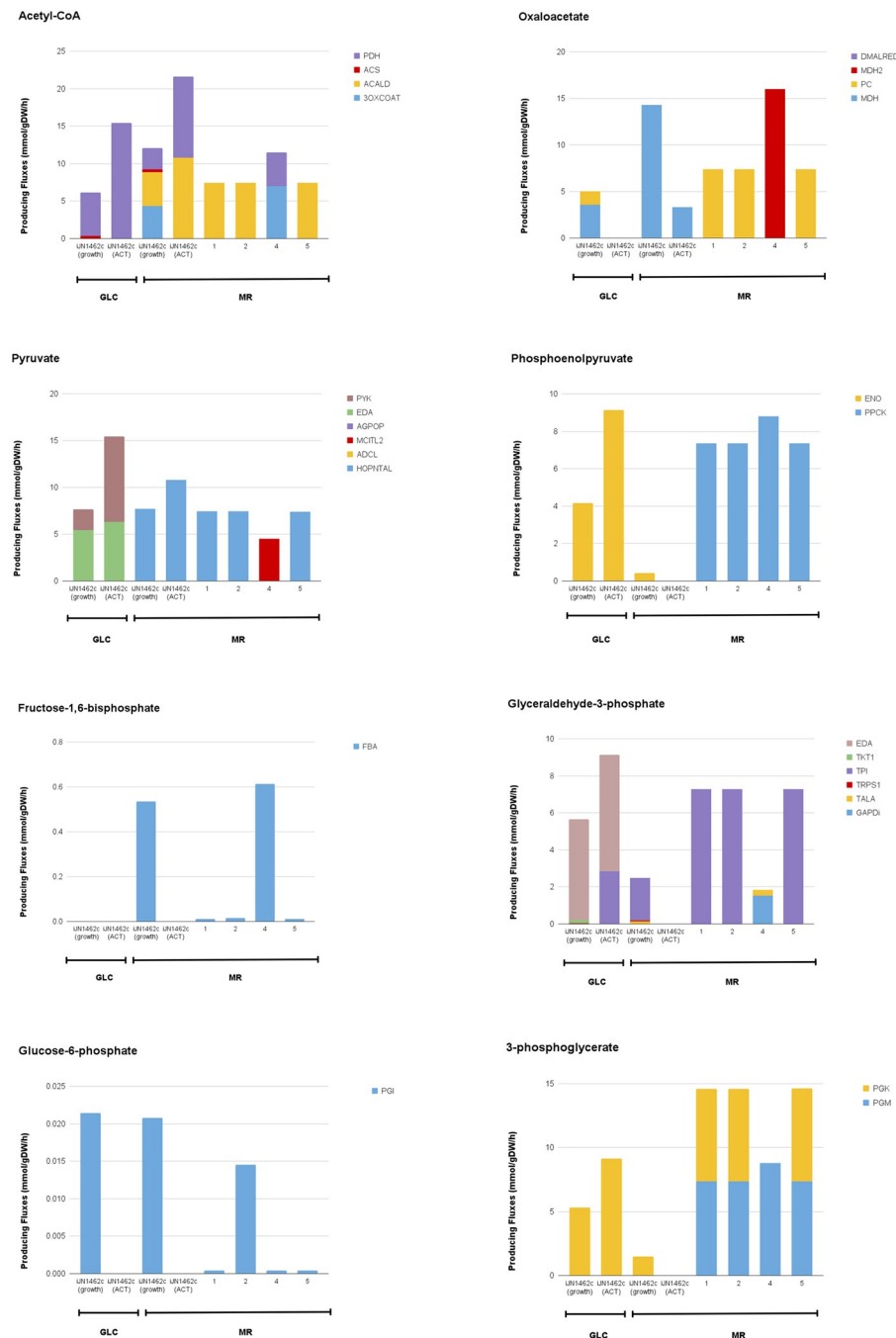

**Fig 5. Flux distributions through key metabolites in lower and upper glycolysis, gluconeogenesis, and the EMP pathway in iJN1462c and suggested growth-coupled strain designs.** Calculated in *in silico* M9 Minimal Medium with MR or GLC as the sole carbon source. iJN1462c was observed twice in different optimization objectives: biomass and ACT production; all growth-coupled designs were observed on MR only and were optimized for biomass/growth.

pathway into dihydroxyacetone phosphate and 3-phospho-D-glyceroyl phosphate by TPI and GAPD respectively. Optimizing ACT production on GLC, on the other hand, requires additional flux through gluconeogenesis via reversal of the TPI reaction.

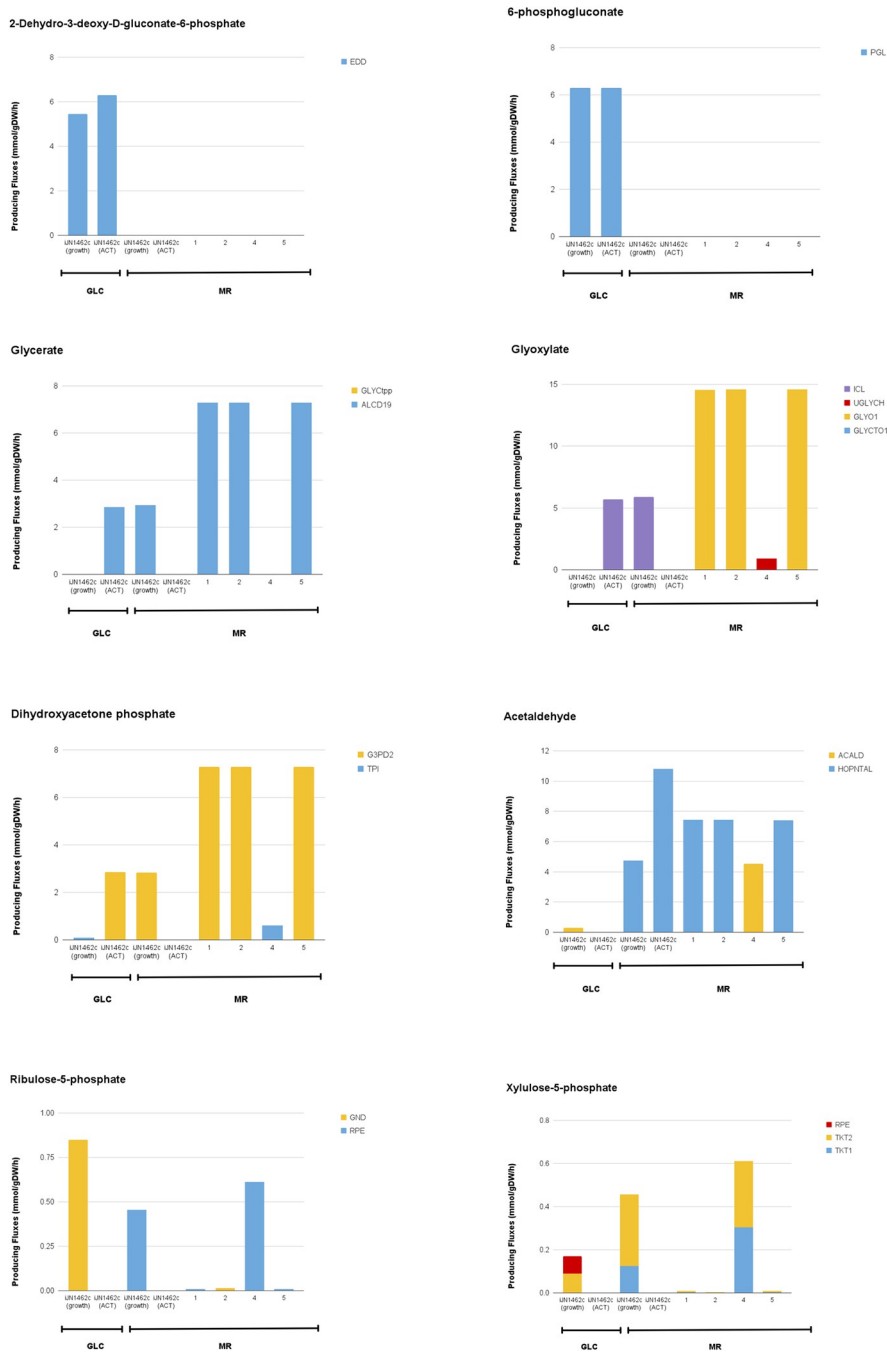

**Fig 6. Flux distributions through key metabolites in the aromatic catabolism pathway (acetaldehyde), ED pathway, PP pathway, and the glyoxylate and glycerate shunt in iJN1462c and suggested growth-coupled strain designs.** Calculated in *in silico* M9 Minimal Medium with MR or GLC as the sole carbon source. iJN1462c was observed twice in different optimization objectives: biomass and ACT production; all growth-coupled designs were observed on MR only and were optimized for biomass/growth.

On MR, activation of gluconeogenesis—and thus the reversal of TPI—is necessary to direct carbon fluxes from aromatic catabolic pathways to the PP pathway (ribulose-5-phosphate and xylulose-5-phosphate) and amino acid biosynthesis (e.g., glutamate, aspartate, serine, leucine)

needed for growth. Similarly, reactions through dihydroxyacetone phosphate and glycerate are reversed when the cell is maximizing ACT production on GLC, and growth on MR, compared to maximizing growth on GLC. Additionally, a high level of PGK flux to generate 3-phospho-glycerate also indicates the activation of gluconeogenesis in these scenarios. Optimization of ACT on GLC and growth on MR are also the only cases where the glyoxylate shunt appears to be activated (ICL). Additionally, for iJN1462c grown on MR, both ICL (**Fig 6**) and ICDHyr (converts isocitrate into a-ketoglutarate) are active at the same time (**Fig 7**). This unusual activation of both the glyoxylate shunt and the TCA cycle occurs because the flux from glyoxylate is used to support flux directed to gluconeogenesis, hence TCA cycle is still needed for energy generation.

In the growth-coupled strain designs, a-ketoglutarate is formed mainly through PSERT, a reaction in serine biosynthesis (**Fig 7**). In these strains, the cell uses flux from gluconeogenesis through the 3-phosphoglycerate node for a-ketoglutarate and serine biosynthesis. Glutamate was also needed as a precursor for PSERT, which is why a high level of glutamate flux was seen following an increase in serine-producing fluxes.

Another unique feature of strain designs 1, 2, and 5 is that metabolites in their TCA cycle (i.e., fumarate, citrate) are produced in low amounts just enough to maintain viability (**Fig 7**). The TCA cycle is a major route to obtain NADH for oxidative phosphorylation, producing a net of 3 NADH molecules per cycle. In strain design 1, 2, and 5, cMCS suggests knockouts that targets reactions in the TCA cycle (e.g., MALS, MDH, SUCDi, SUCOAS, ACONTa) or other reactions that directly or indirectly leads to the production of NADH (e.g., FDH, ALDD2x). This design creates a non-trivial rewiring of fluxes. With the major NADH generating pathways knocked out or tuned down, the cell is now forced to utilize the ACT biosynthesis pathway, to produce the NADH needed for MR degradation and growth. This growth-coupled design can only be feasible with MR as a carbon source because MR is an oxidized molecule that requires NADH in its catabolic pathway, whereas GLC is a reduced molecule which produces NADH via glycolysis. Utilizing MR instead of GLC generates more demand for NADH, and with the knockouts implemented, ACT biosynthesis becomes highly crucial for growth on MR.

The growth-coupled designs further increased NADH demand through other surprising means: restricting NADPH and pyruvate production, both of which are required for growth and ACT biosynthesis. Aside from rewiring NADH production, the knockout targets include reactions that are involved in NADPH production, specifically in the PP pathway, such as PGL, RPE, and ALDD2y. The PP pathway and gluconeogenesis are the two major producers of NADPH. Blocking some routes in the PP pathway rewires NADPH production such that gluconeogenesis becomes the major NADPH synthesis route. For growth-coupling, gluconeo-genesis is preferable to maintain because this pathway pulls flux directly from pyruvate, and with more pyruvate supply needed, the cell is urged to catabolize more MR, creating a larger NADH demand. This explanation is further confirmed by the presence of knockout suggestions that are pyruvate producing reactions (e.g., MICITDr) or reactions that supply flux to pyruvate-derived metabolites (e.g., ACACt2pp, ACACT11). This implies that the cell is enforced to source carbon flux only from MR catabolism for pyruvate synthesis. We also observe an increase in acetaldehyde-producing flux in growth-coupled strain designs (**Fig 6**), indicating more flux coming from MR catabolism in these strains.

The rewiring of metabolism through gluconeogenesis is only possible in MR, because gluconeogenesis is shown to be unnecessary for maximum growth on GLC but is necessary for maximum growth on MR (**Fig 5**). Supporting the observations made by Kukurugya *et al.* [53], on sugar-based substrates, carbon flux flows into the central metabolism via glycolysis and is prioritized for cellular maintenance (ATP and NADH production in glycolysis) before

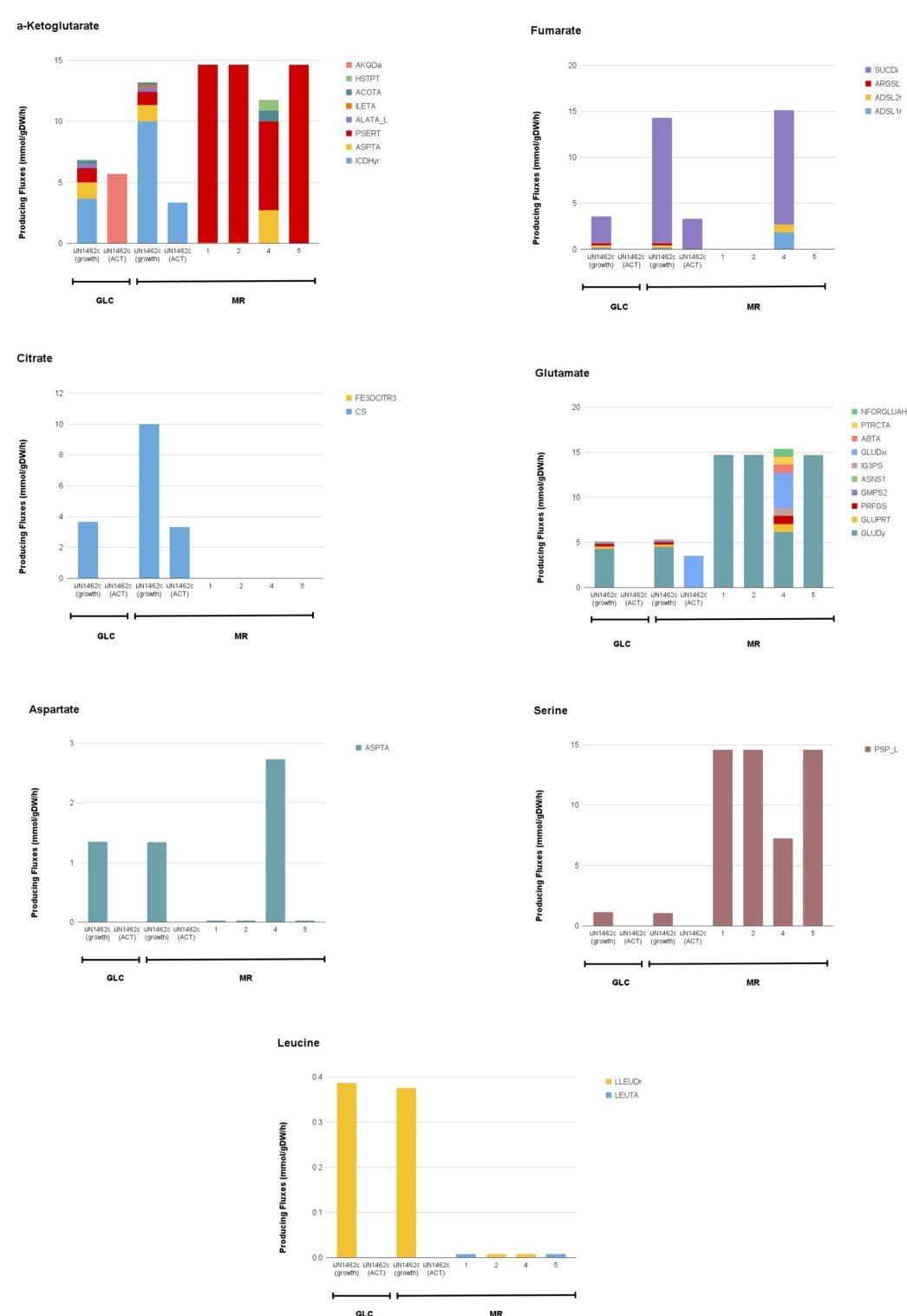

**Fig 7. Flux distributions through key metabolites in the TCA cycle and amino acid biosynthesis pathways in iJN1462c and suggested growth-coupled strain designs.** Calculated in *in silico* M9 Minimal Medium with MR or GLC as the sole carbon source. iJN1462c was observed twice in different optimization objectives: biomass and ACT production; all growth-coupled designs were observed on MR only and were optimized for biomass/growth.

biosynthesis. On aromatic substrates, the flux is directed to acetyl-CoA first and cellular maintenance is managed through gluconeogenesis. Our results suggest that the non-uniform distribution of aromatic versus sugar-based carbon fluxes in *P. putida* is a unique attribute that is advantageous when developing strain designs for growth-coupled production of pigmented secondary metabolites like ACT that generates excess NADH. Growth-coupled regimes that would not otherwise be feasible with the GLC-only medium could be unlocked by using aromatic substrates.

We also expanded our cMCS search on several other aromatic carbons that utilize similar catabolic pathways as MR, namely benzoate, 4-hydroxybenzoate, vanillin, vanillate, ferulate, and gallate (**S3 File**). We found that growth-coupled regimes are only feasible on aromatics that have 0 net NADH or require at least a net of 1 NADH in their catabolism, namely benzoate, 4-hydroxybenzoate, and gallate. Growth-coupling was not possible on vannilate, vannilin, or ferulate. Moreover, we retrieved the flux distributions for the four suggested strain designs found on MR when grown on the six other aromatic substrates (**S6 File**). Although growth-coupling regimes exist for three of the tested aromatic compounds (**S3 File**), we found that the specific growth-coupled phenotype found on MR could not be replicated using the same strain design on other aromatics, further confirming that cMCS suggestions are unique to the carbon source specified prior to the cMCS computation.

## Evaluation of suggested strain designs

For each growth-coupled strain design suggested by cMCS on MR-only and GLC+MR media, we calculated their minimum guaranteed ACT yield, ACT yield at optimum growth, and biomass-product coupled yield (**S3 File**). The results for the four best strain designs (strain designs 1, 2, 4, and 5) are shown in **Table 3** and **Fig 8**.

Flux values used for these calculations were obtained by simulating the strain designs with FBA. Product yield reflects how efficient the system was at converting MR into ACT, while BPCY measures the likelihood that an engineered individual would evolve to its desired phenotype [31]. From our results, we found that strain design 1, 2, and 5 guaranteed the same minimum ACT yield at 0.01145 mmol/gDW/h, while strain design 4 guaranteed a lower minimum. Similarly, strain design 4 also showed the lowest ACT yield when growth is optimized. This difference in ACT yield is likely a result of the different flux distributions we saw in strain design 4 compared to the other suggested designs (**Fig 5**). The design with the highest ACT yield is strain design 5, however, it has a considerably lower BPCY compared to strain design 1 and 2. Since ACT yield is almost similar between strain design 1, 2, and 5, BPCY is worth more consideration when choosing a design to be implemented *in vivo* and optimized through adaptive laboratory evolution. A cell with high BPCY such as strain design 1 is more capable of maintaining an evolutionary advantage due to its higher growth rate than strain design 5. Another factor that should be considered is the number of genetic modifications needed to

**Table 3. Flux distributions, yields, and BPCY of the strain designs.**

| Strain Design | Substrate Uptake (mmol/gDW/h) | Biomass (mmol/gDW/h) | ACT production (mmol/gDW/h) | Min. Guaranteed Yield | ACT Yield at Optimum Growth |
|---|---|---|---|---|---|
| iJN1462c | 9.176 | 0.5976 | 0.000 | 0.000 | 0.000 |
| 1 | 7.431 | 0.01270 | 0.4599 | 0.01145 | 0.06190 |
| 2 | 7.428 | 0.01235 | 0.4599 | 0.01145 | 0.06192 |
| 4 | 7.018 | 0.01231 | 0.3214 | 0.002726 | 0.04579 |
| 5 | 7.429 | 0.01162 | 0.4611 | 0.01145 | 0.06208 |

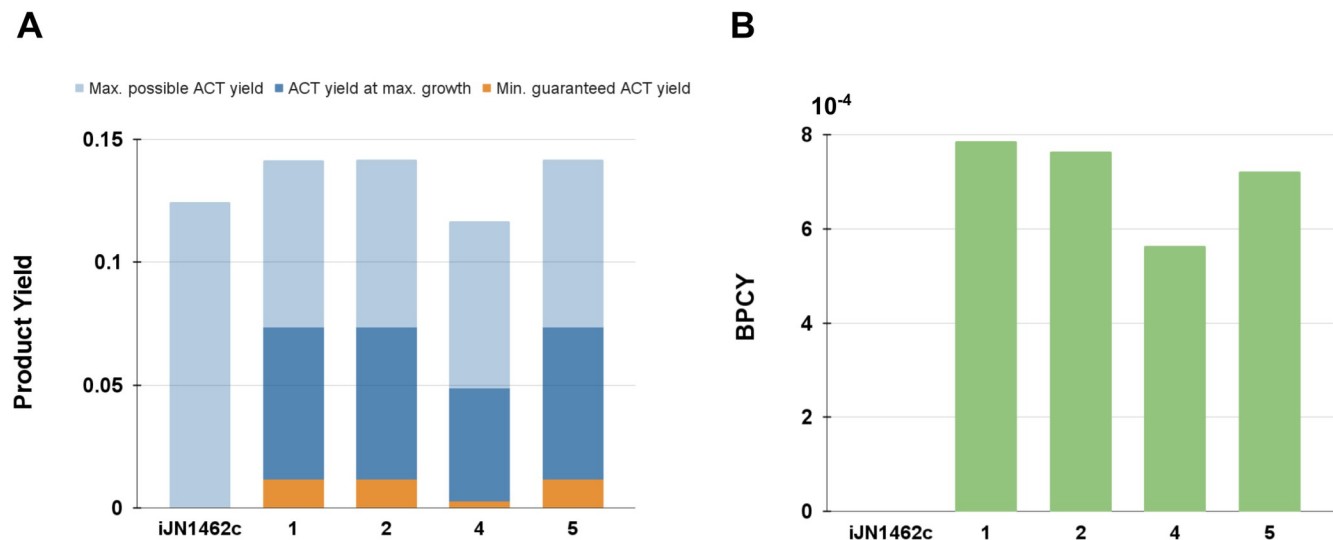

**Fig 8. Product yield and biomass-product coupled yield (BPCY) of the strain designs suggested by cMCS.** (A) Minimum guaranteed ACT yield (orange), ACT yield when BIOMASS_KT2440_WT3 is optimized (dark blue), and maximum theoretical ACT yield (light blue) of suggested strain designs compared to iJN1462c. (B) BPCY values of the strain designs compared to iJN1462c at the exponential phase.

actualize the strain design. Strain design 1 and 2 require only 12 knockouts while strain design 5 requires 17 knockouts. These number of knockouts or knockdowns are possible as already attempted by others with a multiplex CRISPR system or other genome engineering methods [29,54,55].

## Conclusion

Industrial wastes such as azo dyes have the potential to be reused as feedstock for biomanufacturing high-value compounds and secondary metabolites. With the development of various azo dye separation processes from wastewater through adsorption, coagulation, and membrane filtration [56,57], azo dyes have become more accessible for microbial systems to degrade and valorize into new compounds. Through this work, we have demonstrated that it is plausible to implement a system in *P. putida* KT2440 to produce ACT from MR. Although FBA/FVA analyses revealed flux competitions between dye degradation, growth, and ACT production, we consider aromatic compounds, particularly azo dyes, to be an incredibly interesting feedstock to explore for a polyketide biosynthesis because using MR as a carbon source allows us to access growth-coupling regimes for ACT that could not be realized with growth on GLC. Moreover, utilizing MR exhibited less ATP turnover while generating a maximum theoretical ACT production flux that was 18.28% higher than using GLC. The growth-coupled strain designs and metabolic interventions presented here sets the stage for further experimentation and improvement *in vivo*. Our workflow provides a useful framework that might be adapted to produce other important pigmented polyketides or industrial chemicals from azo dyes. For example, one could potentially apply the same approach to growth-couple and produce acetate, acetoin, or 2,3-butanediol [58] from MR which, although not a polyketide, also generate NADH in its biosynthesis similar to ACT. Furthermore, we have shown that the selection of appropriate carbon sources can alter the growth-coupling solution spaces for a target metabolite, consequently restricting or expanding possible metabolic engineering approaches that can be implemented.

## Supporting information

**S1 File. iJN1462c metabolic model.** Modified iJN1462 *P. putida* KT2440 model containing the azo dye to actinorhodin synthetic pathway.
(XML)

**S2 File. Summary of cofactor flux distributions in iJN1462c.** Contains a list of cofactor consuming and producing reactions along with their fluxes retrieved through FBA/FVA.
(XLSX)

**S3 File. cMCS computation results.** Includes results of cMCS computations on GLC, MR, GLC + MR and six other aromatic substrates.
(XLSX)

**S4 File. OptKnock, OptGene, and cMCS strain design suggestions.** Shows the production envelopes of strain designs generated by OptKnock, OptGene, and cMCS.
(DOCX)

**S5 File. Reactions involved in the azo dye to actinorhodin synthetic pathway.** Contains all the native and non-native reactions added to iJN1462c to enable conversion of methyl red to actinorhodin.
(DOCX)

**S6 File. Simulation on other aromatic carbon sources.** Evaluation of the strain designs found by the cMCS computation on MR in other common aromatic carbon sources.
(DOCX)

## Acknowledgments

The authors would like to thank the National Research and Innovation Agency of the Republic of Indonesia (BRIN) for organizing the National Youth Scientific Competition (Lomba Karya Ilmiah Remaja Nasional) 2021 and facilitating the research mentoring program for junior and senior high school students, thus, this study can be conducted.

## Author Contributions

**Conceptualization:** Parsa Nayyara, Dani Permana, Ratih Fahayana.

**Data curation:** Parsa Nayyara.

**Formal analysis:** Parsa Nayyara.

**Funding acquisition:** Dani Permana.

**Investigation:** Parsa Nayyara.

**Methodology:** Parsa Nayyara, Dani Permana.

**Project administration:** Dani Permana.

**Resources:** Dani Permana.

**Software:** Parsa Nayyara.

**Supervision:** Dani Permana, Riksfardini A. Ermawar, Ratih Fahayana.

**Validation:** Dani Permana.

**Visualization:** Dani Permana.

**Writing – original draft:** Parsa Nayyara.

**Writing – review & editing:** Dani Permana, Riksfardini A. Ermawar, Ratih Fahayana.

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
