## [Decision Letter · Decision Letter 0]

29 Aug 2023

PONE-D-23-23251Computational Analysis into the Potential of Azo Dyes as a Feedstock for Actinorhodin Biosynthesis in Pseudomonas putidaPLOS ONE

Dear Dr. Permana,

Thank you for submitting your manuscript to PLOS ONE. After careful consideration, we feel that it has merit but does not fully meet PLOS ONE’s publication criteria as it currently stands. Therefore, we invite you to submit a revised version of the manuscript that addresses the points raised during the review process.

While the reviewers acknowledge that the manuscript presents an interesting piece of work, there are many concerns that need to addressed carefully before the manuscript is deemed suitable for publication. I request the authors to carefully revise the manuscript taking into consideration all of the reviewers' concerns. Please submit your revised manuscript by Oct 13 2023 11:59PM. If you will need more time than this to complete your revisions, please reply to this message or contact the journal office at plosone@plos.org. Please include the following items when submitting your revised manuscript:A rebuttal letter that responds to each point raised by the academic editor and reviewer(s). You should upload this letter as a separate file labeled 'Response to Reviewers'.A marked-up copy of your manuscript that highlights changes made to the original version. You should upload this as a separate file labeled 'Revised Manuscript with Track Changes'.An unmarked version of your revised paper without tracked changes. You should upload this as a separate file labeled 'Manuscript'.

We look forward to receiving your revised manuscript.

Kind regards,

Karthik Raman, Ph.D.

Academic Editor

PLOS ONE

Journal Requirements:

"The author(s) received no specific funding for this work. No funding sources are to be declared as this study was a part of the National Youth Scientific Competition (Lomba Karya Ilmiah Remaja Nasional) 2021 which was organized and supported by the National Research and Innovation Agency of the Republic of Indonesia (BRIN). "

6. Please upload a new copy of Figure 1 as the detail is not clear. Please follow the link for more information: " ext-link-type="uri" xlink:type="simple">https://blogs.plos.org/plos/2019/06/looking-good-tips-for-creating-your-plos-figures-graphics/"
" ext-link-type="uri" xlink:type="simple">https://blogs.plos.org/plos/2019/06/looking-good-tips-for-creating-your-plos-figures-graphics/"

7. We are unable to open your Supporting Information file [Supporting Information 1]. Please kindly revise as necessary and re-upload.

Reviewers' comments:

Reviewer's Responses to Questions

**Comments to the Author**

1. Is the manuscript technically sound, and do the data support the conclusions?

Reviewer #1: Partly

Reviewer #2: Yes

2. Has the statistical analysis been performed appropriately and rigorously? 

Reviewer #1: N/A

Reviewer #2: Yes

3. Have the authors made all data underlying the findings in their manuscript fully available?

Reviewer #1: Yes

Reviewer #2: No

4. Is the manuscript presented in an intelligible fashion and written in standard English?

Reviewer #1: Yes

Reviewer #2: Yes

5. Review Comments to the Author

Reviewer #1: The manuscript by Nayyara and coworkers explores the potential of using azo dye methyl red as the sole carbon and energy source for growth and couple it to production of a polyketide in the gram negative soil bacterium P.putida and enumerate strain designs for the same using constrained minimal cut sets. While this is theoretically feasible and the simulations quite exhaustive, the lack of a preliminary experimental confirmation to ascertain the potential of the host to use the said azo dye in a cofactor intensive catabolic pathway as a sole carbon source needs to be verified and/or such experimental growth data cross-verified with the simulations (in the wild-type model) before ambitiously pairing it to a polyketide production. The results section was difficult to follow as well due to a lack of comparative/superimposed flux maps for the strain designs. The manuscript can be restructured to make it easily decipherable for the readers. Considering the heavy engineering that needs to go into such a growth coupled strain (17 gene knockouts or knockdowns) on top of the metabolic burden enforced by the substrate (toxic above a certain threshold) and multistep pathway-based target metabolite production (either on a replicative plasmid or genomically integrated), it might be prudent to demonstrate growth coupling strategies for other industrially important metabolites with simpler pathways. Overall, I would find this manuscript more compelling with some experimental validations.

Reviewer #2: The article employs flux balance analysis to showcase the promising potential of Azo dyes as an alternative feedstock compared to the commonly employed glucose-rich feedstocks in the production of polyketides. The production envelopes reveal that Methyl red (MR) surpasses glucose as a preferred feedstock for Actinorhodin (ACT) production. This preference is attributed to its lower demand for NADH flux in oxidative phosphorylation, resulting in reduced ATP turnover.

In the context of strain design, the authors looked for different gene knockouts to optimize the synthesis of ACT from MR, utilizing a constrained Minimal Cut Set (cMCS) approach for metabolic pathway analysis. This analysis indicates that the cMCS solutions are tailored to specific carbon sources.

Major comments

1. No hyperlinks or references are available for the updated metabolic model iJN1462c.

2. Supporting information for the data used in the study is available upon request.

3. There’s inconsistency in reported total number of MCS solutions for different biomass and ACT flux distributions, Page 22 line 1 mentions 203 solutions whereas line 3, mentions 213 MCS solutions.

Minor Comments

1. Pseudomonas putida should be italicized or underlined according to taxonomic nomenclature (Page 2, line 5) (Abstract).

2. Table 1 would benefit from additional information about the reactions (Page 7).

3. Prior to discussing iJN1462c, it's important to include information about iJN1462b (Page 7, line 5).

4. To facilitate the calculation of the rate of cofactor regeneration using the flux sum concept, it would be helpful to include notations for various variables (Page 9, Equation 1).

5. The source and calculation method for the MR uptake rate mentioned in Line 9 of Page 17 have not been provided.

6. The rationale for selecting only four design strains due to reasonable knockout size is missing; an interpretation of this should be included (Page 22)

In closing, this article delves into the potential of Azo dyes, with Methyl red as an alternative feedstock for polyketide production, particularly Actinorhodin. It could pave the way for non-sugar-based substrates to be used in the production of polyketides and other industrially important compounds.

While the research methodology is sound, the manuscript requires some essential refinements. The manuscript requires revision to include all the omitted information and rectify minor errors.

6. PLOS authors have the option to publish the peer review history of their article (what does this mean?). If published, this will include your full peer review and any attached files.

Reviewer #1: No

Reviewer #2: No

---

## [Author Response · Author response to Decision Letter 0]

22 Oct 2023

Dear Dr. Raman,

Thank you for your e-mail dated August 29 regarding our manuscript titled “Computational Analysis into the Potential of Azo Dyes as a Feedstock for Actinorhodin Biosynthesis in Pseudomonas putida” (PONE-D-23-23251) and the attached comments by the two reviewers. We are grateful for the valuable comments made by the reviewers.

We have carefully revised the manuscript in accordance with the comments and suggestions raised. We apologize for our slow response to address questions and comments provided by the reviewers, but we require this time to perform the suggested additional design and visualization of some figures. The revised text is highlighted in green in the manuscript. We attach the revised manuscript, the revised manuscript with track changes, and detailed point-by-point responses to the comments raised by the reviewers that describe the changes we have made. We want to confirm that we deleted the financial disclosure and funding information because we did not receive any funding for our study. We also revised the statement about the data availability. Regarding Supporting Information 1, because the file type is XML, we just uploaded it as it is. We believe that the manuscript has been improved satisfactorily and hope that the revised version is now acceptable for publication in PLOS One. 

Once more, we greatly appreciate the valuable comments provided.

We look forward to hearing from you at your earliest convenience.

Yours sincerely,

Dani Permana, Ph.D.

---

## [Decision Letter · Decision Letter 1]

7 Dec 2023

PONE-D-23-23251R1Computational Analysis into the Potential of Azo Dyes as a Feedstock for Actinorhodin Biosynthesis in Pseudomonas putidaPLOS ONE

Dear Dr. Permana,

Thank you for submitting your manuscript to PLOS ONE. The reviews for your manuscript are now in and are quite positive. Please revise your manuscript taking into account the comments of both reviewers. I do understand that experiments may not be readily doable, but do consider the constructive suggestions of both reviewers to strengthen your manuscript further.

We look forward to receiving your revised manuscript.

Kind regards,

Karthik Raman, Ph.D.

Academic Editor

PLOS ONE

Journal Requirements:

Reviewers' comments:

Reviewer's Responses to Questions

**Comments to the Author**

1. If the authors have adequately addressed your comments raised in a previous round of review and you feel that this manuscript is now acceptable for publication, you may indicate that here to bypass the “Comments to the Author” section, enter your conflict of interest statement in the “Confidential to Editor” section, and submit your "Accept" recommendation.

Reviewer #1: (No Response)

Reviewer #2: All comments have been addressed

2. Is the manuscript technically sound, and do the data support the conclusions?

Reviewer #1: Yes

Reviewer #2: Yes

3. Has the statistical analysis been performed appropriately and rigorously? 

Reviewer #1: N/A

Reviewer #2: N/A

4. Have the authors made all data underlying the findings in their manuscript fully available?

Reviewer #1: No

Reviewer #2: Yes

5. Is the manuscript presented in an intelligible fashion and written in standard English?

Reviewer #1: Yes

Reviewer #2: Yes

6. Review Comments to the Author

Reviewer #1: The authors have written a well thought-out response with necessary changes that considerably strengthens their hypothesis and enhances the flow of the manuscript. I do agree that this article is a nice demonstration of the effect of non-conventional carbon sources on target chemical production using genome scale models. Some of these nice rebuttal statements would actually make for a good conclusion or add-ons in the manuscript. Thanks to the authors for adding necessary references (for degradation of methyl red by P.putida) and the much improved and now legible Fig.4.

Specific comments:

1. I still standby my statement that an experimental demonstration would enhance the appeal of the work (and with a complete understanding of the limited access to funding and resources). By experimental demonstration - I do not mean the implementation of the time and labor-intensive growth coupling designs, rather the MR utilization by KT2440 as the sole carbon source is very vital to this study. The newly added citations are great examples for MR decolorization (maybe to a colorless product) but not utilization as a sole carbon source (those cells were grown on rich medium and MR already shows toxicity on rich medium). Going by the authors’ statements one might (worst case) or might not (best case) even need to express AnthDO to carry out this growth study. Therefore, I leave it to the editor’s discretion as to whether this is necessary for supporting the author’s claims. A recent and a relevant product substrate paired study using an aromatic compound from feedstocks (using p-coumarate as the sole carbon source) to Indigoidine by Eng et al., 2023 (Cell reports) in P.putida leaves room for discussion that even a 3 gene cutset (plus only a 2-step heterologous pathway) needed lab evolution to improve growth even on a native aromatic carbon source.

2. Since 2 of the 4 chosen designs talk about the meta-pathway for degradation and you mention KT2440 in the manuscript, this particular strain (not the in silico model) is a TOL plasmid free strain (Nelson et al., 2002) and the genes for the meta-cleavage pathway for catechol degradation is located on the TOL plasmid. It would be prudent to suggest that any interested reader has to express these additional genes when they actually do the experiment. Also, as rightly pointed out by the authors, the ortho-cleavage pathway is highly regulated and although the model chooses the “shortest” or thermodynamically favorable path to the product according to the design interventions- it will not be the case in an actual cell where you might have to additionally delete the competing pathways for uptake/metabolism (the model has no deletion targets in these parallel pathways).

3. Growth-coupling can actually be difficult to demonstrate for even routine secondary metabolites on different carbon sources. Thanks for the clarification on the choice of ACT. I do agree that it is a valuable and a suitable target (given its NADH production capability that seems to be the critical point to drive MR utilization and coupling with endogenous NADH generators knocked out). Whether it would pose a challenge to actually implement them (22 step ACT pathway +the many knockouts of the design + expression of the meta-cleavage pathway genes in KT2440) and discourage one from implementation (again for lack of funding/resources), I shall refrain from commenting at this stage. Could you perhaps add in a few more products in the discussion that are easier to implement (you need not validate but which might work using a similar mechanism?) if you find it suitable? This will be a food for thought to the readers and will motivate them to pursue this line of experiments.

4. The model- is iJN1462 the latest model or 63? Also, could you please add in the reactions through the peripheral pathways (glucose to gluconate etc.) in Fig 5? What were the fluxes through them (apart from ED-EMP) on glucose +MR?

5. ATP maintenance reaction- Perhaps I was not clear enough. I was wondering if the reaction for ATP maintenance in the model was constrained to an experimental value or unconstrained?

6. Section 2.5- From your response, I believe there was only one reaction “AnthDo” that was used for gap-filling? The description of the process in the methods section though seems like it went through a lot of iterations for a lot of reactions. Could you perhaps rephrase it?

7. Page-14 and line 10: Thanks for the clarification. Could you also if the same applies to iJN1462a (without Act reactions as well)?

8. Section 2.7: Although the authors might find it redundant or trivial, I would recommend that they atleast make their codes (if not the details of each step) for everything available to readers for reproducibility. Apologies if it was included already in the supplementary.

9. On glc+MR medium or in a MR only medium (say in a fed-batch process), do you expect anthranilate to get converted to "trp" once serine becomes available through biosynthesis (not in silico but as a point of discussion for an actual scenario)? I feel that this alternate degradation pathway should be mentioned in the text as well if not already done. Your simulations in the response was pretty convincing and this fact could also advocate for a carbon specific growth coupling despite a competing pathway's presence.

Reviewer #2: The authors have revised the manuscript, incorporating the reviewers' suggestions and comments. The manuscript now presents a clear and comprehensive explanation, making it more accessible to a broader audience. The inclusion of metabolic maps and detailed tables with annotated equations further improves the clarity. With the aim of further enhancing the manuscript, I would like to propose the following suggestions:

1. To further highlight the novelty of your work, which considers different flux spaces with non-sugar based sources, please provide examples and references for criteria of selecting carbon sources which does not takes the solution space for the specific metabolite into account.

2. Could you provide a brief explanation for focusing solely on the meta-cleavage pathway? Given that both meta and ortho cleavage pathways catabolize catechol to acetyl-CoA, a common entry point for the TCA cycle and fatty acid biosynthesis, this explanation would help readers better understand your decision-making process. (Page 16)

3. Would you be willing to provide a rationale or derivation for the threshold values of MR uptake rate? This will provide additional context and support for the derived values. (Page 19, line 3)

While your work provides valuable insights, it will surely benefit from the addressing suggestions

provided above.

7. PLOS authors have the option to publish the peer review history of their article (what does this mean?). If published, this will include your full peer review and any attached files.

Reviewer #1: No

Reviewer #2: No

---

## [Author Response · Author response to Decision Letter 1]

4 Feb 2024

We express our appreciation for the valuable comments raised by the reviewers from first and second round. We have carefully revised the manuscript in accordance with their suggestions. We have attached a detailed point-by-point response to the comments raised by each reviewer. These responses describe the changes we made to the manuscript. We believe that the revised manuscript has been improved satisfactorily and hope that this version is now acceptable for publication in PLOS ONE.

---

## [Editor Report · Decision Letter 2]

6 Feb 2024

Computational Analysis into the Potential of Azo Dyes as a Feedstock for Actinorhodin Biosynthesis in Pseudomonas putida

PONE-D-23-23251R2

Dear Dr. Permana,

We’re pleased to inform you that your manuscript has been judged scientifically suitable for publication and will be formally accepted for publication once it meets all outstanding technical requirements.

Kind regards,

Karthik Raman, Ph.D.

Academic Editor

PLOS ONE

Additional Editor Comments (optional):

The authors have satisfactorily addressed all reviewer concerns.

---

## [Editor Report · Acceptance letter]

22 Feb 2024

PONE-D-23-23251R2 

PLOS ONE

Dear Dr. Permana, 

I'm pleased to inform you that your manuscript has been deemed suitable for publication in PLOS ONE. Congratulations! Your manuscript is now being handed over to our production team.

Kind regards, 

on behalf of

Dr. Karthik Raman 

Academic Editor

PLOS ONE